# Effects of Low-Protein Amino Acid-Balanced Diets and *Astragalus* Polysaccharides on Production Performance, Antioxidants, Immunity, and Lipid Metabolism in Heat-Stressed Laying Hens

**DOI:** 10.3390/ani15162385

**Published:** 2025-08-14

**Authors:** Wenfeng Liu, Xiaoli Wan, Zhiyue Wang, Haiming Yang

**Affiliations:** College of Animal Science and Technology, Yangzhou University, Yangzhou 225009, China; dx120220169@stu.yzu.edu.cn (W.L.); wanxl@yzu.edu.cn (X.W.); dkwzy@263.net (Z.W.)

**Keywords:** oxidative stress, follicular development, antioxidant enzymes, immunomodulation, anti-inflammatory cytokine

## Abstract

This study explores how low-protein amino acid-balanced (LPAB) diets with *Astragalus* polysaccharides (APSs) influence laying hens under heat stress. The results show that APSs and LPAB diets can mitigate the adverse effects of heat stress on hens. APSs mainly regulate the secretion of gonadotropin by reducing the concentration of corticosterone in serum, thereby increasing the concentration of follicle-stimulating hormone (FSH) and FSH receptor (FSHR) mRNA expression to regulate primary follicle development and increase the egg production rate. While low-protein diets improve yolk color and boost FSH, they also decrease immunoglobulin and triglyceride levels. Adding APSs enhances immunoglobulin concentrations and helps recover cholesterol levels. LPAB diets and APSs can both enhance cholesterol uptake and suppress its synthesis. Overall, LPAB diets and APSs are deemed valuable feeding strategies for improving production performance and alleviating heat stress in laying hens.

## 1. Introduction

Rising ambient temperatures beyond the thermoneutral zone are a primary cause of heat stress in poultry, impairing performance [1], while in southern China, summer heat readily triggers acute or chronic stress in laying hens because they lack sweat glands and have dense plumage [2,3]. Heat stress directly damages proteins and indirectly generates ROS [4]. High temperatures also bring about adverse effects such as reduced feed conversion ratio (FCR), feed intake, body weight, egg production, egg weight, and egg quality, which can lead to severe egg loss [5,6]. In addition, high-temperature environments can lead to an increase in plasma corticosterone (CORT) concentrations and affect humoral and cellular immune responses, potentially leading to immune dysfunction [7,8]. Moreover, a study has shown that heat stress impairs leukocyte protein synthesis, lowers plasma immunoglobulin G (IgG), immunoglobulin M (IgM), and anti-inflammatory cytokine concentrations, and increases the levels of pro-inflammatory cytokines [9].

To counter high temperatures, farmers often use wet curtain cooling and mechanical ventilation. However, these temperature control systems are expensive and hard to maintain consistently in large or open-sided poultry houses. Crude protein has a higher thermic effect because it generates a greater caloric increment [10]. Reducing protein levels in laying hen diets and supplementing with synthetic amino acids to meet essential amino acid (EAA) needs may be a better strategy. Studies have shown that reducing dietary protein levels from 16.5% to 12% under heat stress conditions and supplementing EAAs can improve the resistance of laying hens to heat stress while maintaining laying performance [11]. Dietary supplementation of methionine and L-lysine can improve protein utilization and production performance [12,13].

In addition, *Astragalus* polysaccharides (APSs) play a role due to their anti-inflammatory, immune stimulation, and antioxidant effects [14,15]. In the APS group, the serum levels of tumor necrosis factor-α (TNF-α), interleukin-1β (IL-1β), and interleukin-6 (IL-6) were significantly decreased, and the expression of related genes was also down-regulated [16].

This study investigates the effects of APSs on the production performance, antioxidants, immunity, and lipid metabolism in heat-stressed laying hens based on a low-protein amino acid-balanced diet to provide a reference. This helps to develop a heat stress mitigation strategy that is more suitable for laying hen production systems.

## 2. Material and Methods

### 2.1. Experimental Design and Diets

In the same hen house, 768 Hy-Line Brown laying hens (52 weeks old), with nearly the same body weight, were randomly divided into 4 experimental groups, with 8 replicates in each group and 24 hens in each replicate. Laying hens were observed for 10 days to record individual egg production, feed intake, and health status. The experiment was conducted when the hens were between 52 and 58 weeks of age. According to the egg production, individuals with low egg production or seemingly weak egg production were excluded, and then, the remaining hens were randomly divided into groups. The control group (CON) was kept at a constant 24 ± 1 °C, and the treatment groups (TRT) were exposed to 32 ± 1 °C for 7 h per day (heat stress). Experimental laying hens were from Jurong Haoyuan Ecological Agriculture Technology Co., Ltd. (Zhenjiang, China). The control diet based on maize–soybean meal was formulated to supply the nutrient requirements of the birds [17] (Table 1); the crude protein (CP) level in the basal diet was 15.85%, and in the low-protein diet, it was 13.85%. The CON was fed with basal diet, while the TRT groups were fed with basal diet (HB), low-protein amino acid-balanced (LPAB) diet (HL), and LPAB diets supplemented with 0.5% APSs (HLA). During the experimental period, the hens had unrestricted access to feed and water under a lighting programme of 16 h light: 8 h dark. The chicken cage was a three-layer cage with a length of 40 cm × width of 37 cm × height of 37 cm. Each cage was equipped with a trough and two automatic water dispensers. The diets were prepared, respectively, and all the diets used the same batch of corn, soybean meal and other ingredients. The test additive was APSs powder, purity 98% (*w*/*w*), supplied by Xi’an Shennong Technology Co., Ltd. (Xi’an, China). The polysaccharide was extracted from dried *Astragalus membranaceus* roots by combined ethanol–water extraction, followed by deproteinization, decoloration and lyophilization, yielding a water-soluble, off-white powder. Immediately before diet preparation, APSs were first pre-blended with ten times its weight of corn meal to ensure homogeneity, then incorporated into the basal mash at 0.5% (*w*/*w*) using a horizontal ribbon mixer (5 min mixing time). Heat stress was applied in climate-controlled rooms. From 09:00 to 16:00 each day, temperature was maintained at 32 ± 1 °C and relative humidity at 65 ± 5% using a PID-regulated combination of electric heaters and a variable-speed wet-curtain system; outside this period the rooms were held at 24 ± 1 °C.

### 2.2. Animals and Sample Collection

During the experiment, the total egg production, total egg weight, feed intake and mortality number were recorded every day, and the average egg weight, egg production rate, breakage rate (broken egg number/total egg number), mortality rate, average daily feed intake and FCR were counted with repetition as the unit. Three eggs were randomly selected from each replicate on day 40, day 41 and day 42 for egg quality analysis. Eggshell strength was measured by Egg Force Reader (Orka Technology Ltd., West Bountiful, UT, USA). Eggshell thickness was measured by Egg Force Shell Thickness (Orka Technology Ltd., West Bountiful, UT, USA). The egg shape index was calculated as the vertical diameter divided by the horizontal diameter, following the NY/T 823-2020 method [18]. Egg yolk weight and egg weight were measured by electronic balance (precision, 0.01 g) to calculate the relative egg weight. The egg yolk color, albumen height and Haugh unit were measured by Egg Nnalyzer^TM^ (Orka Technology Ltd., West Bountiful, UT, USA). On day 42 of the experimental period, 24 healthy Hy-Line laying hens were randomly selected from each treatment, and blood samples were taken from their brachial veins. The blood samples were centrifuged at 826× *g* for 10 min to prepare serum, which was then divided into enzyme-free tubes and stored in a −20 °C freezer. Experimental laying hens were euthanized after cervical dislocation, the ovary and liver were removed after death, the ovary was rinsed with cold PBS, and the number of follicles in the ovaries was determined by diameter, small white follicle (SWF, 2–4 mm), big yellow follicle (BYF, 4–10 mm), graded follicle (GF, diameter > 10 mm) and the number of follicles in the ovaries was counted [19], and all visible hierarchical and pre-hierarchical follicles were carefully removed. The remaining ovarian tissue (excluding the follicles) was then rapidly frozen in liquid nitrogen and stored at −80 °C until RNA extraction for determining the mRNA expression of follicle-stimulating hormone receptor (*FSHR*), luteinizing hormone receptor (*LHR*), estrogen receptor 1 (*ESR1*), estrogen receptor 2 (*ESR2*), β-actin, superoxide dismutase (*SOD1*), nuclear factor erythroid 2-related factor 2 (*Nrf2*), glutathione peroxidase (*GPx*), tumor necrosis factor-α (*TNF-α*), IL-1β, interleukin-10 (*IL-10*), interferon-γ (*IFN-γ*), sterol regulatory element-binding protein (*SREBP2*), 3-hydroxy-3-methylglutaryl-CoA (*HMGCR*), low-density lipoprotein receptor (*LDLR*).

### 2.3. Measurement of Serum Hormone, Antioxidant, Immunoglobulin, Inflammatory and Biochemical Indicators

Serum concentration of adrenocorticotropic hormone (ACTH), CORT, estradiol (E_2_), FSH, LH and progesterone (P_4_) were measured using a one-step sandwich ELISA with a double antibody kit from Elabscience (Houston, TX, USA). Serum concentration of immunoglobulin A (IgA), IgG, and IgM, inflammatory factors IL-1β, interleukin-6 (IL-6), IL-10, monocyte chemokine-1 (MCP-1), IFN-γ, and tumor necrosis factor-α (TNF-α) were measured using ELISA kits purchased from Shanghai Yubo Biotechnology Co., Ltd. (Shanghai, China). The total protein concentration in the serum was measured using a kit, and the activities of total SOD (T-SOD), GPx, total antioxidant capacity (T-AOC), and malondialdehyde (MDA) were measured using kits purchased from Nanjing Jiancheng Bioengineering Institute (Nanjing, China). Total protein (TP), albumin (ALB), globulin (GLB), serum total cholesterol (T-CHO), high-density lipoprotein cholesterol (HDL-C), LDL cholesterol (LDL-C), triglyceride (TG) and glucose (GLU) content, as well as the activity of aspartate aminotransferase (AST) and alanine aminotransferase (ALT) were measured in the serum using a UniCel DXC 800 Synchron fully automated biochemical analyzer (Beckman Coulter Inc., Brea, CA, USA).

### 2.4. Expression of Gonadotropin-Releasing Hormone mRNA in the Liver and Ovaries

Total RNA isolation, cDNA synthesis, and quantitative PCR were performed according to the methods described in a previously published paper [20]. The specificity of the primers was confirmed by melting curve analysis. The reference gene was *β-actin*, and the primer sequences for each gene are listed in Table 2. The 2^−ΔΔCt^ method was used to calculate the relative expression levels of the target genes.

### 2.5. Statistical Analysis

Microsoft Excel 2019 (v. 1808, Microsoft, Redmond, WA, USA) was used for preliminary data processing. Statistical analysis was conducted using SPSS software, version 19.0 (SPSS Inc., Chicago, IL, USA). Prior to all parametric tests, the homogeneity of variances was assessed using Levene’s F-test. One-way analysis of variance (ANOVA) was performed to compare differences among the four experimental groups (CON, HB, HL, and HLA), followed by post hoc multiple comparisons using Tukey’s test. The experimental design was not a fully factorial arrangement. Specifically, the LPAB and APS treatments were only applied under heat stress conditions (32 °C), while the thermoneutral group (24 °C) received only the basal diet. Therefore, temperature, protein level, and APS supplementation were not treated as independent factors in a multifactorial model. Data were presented as mean and pooled standard error of the mean (SEM), and statistical significance was set at *p* < 0.05.

## 3. Results

### 3.1. Production Performance and Follicle Development

As presented in Table 3 and Table 4, compared with the CON, the breakage rate, mortality, average daily feed intake and average egg weight of HB, HL and HLA were not significantly different. In terms of egg production rate (EPR), the CON was the highest, and HB and HL were significantly decreased (*p* < 0.05). Compared with CON, HL and HLA, the FCR of HB worsened significantly (*p* < 0.05). The number of SWF in the CON, HL and HLA was more than HB (*p* < 0.05).

### 3.2. Egg Quality and Egg Components

According to Table 5, compared to the CON, the LPAB diets deepened yolk color (*p* < 0.05). Temperature, protein level in the diets and APS addition had no significant effect on albumen height, Haugh unit, yolk ratio, eggshell strength, eggshell thickness and egg shape index (*p* > 0.05).

### 3.3. Serum Hormone Indicators and Relevant Gene Expression in the Ovary

According to Table 6, the HB had higher concentration of ACTH, CORT, whereas FSH was lower compared to the other groups (*p* < 0.05). Additionally, the HL had higher concentration of CORT compared to CON and HLA (*p* < 0.05). Figure 1 demonstrated that the *FSHR* in the ovary was up-regulated in CON, HL and HLA (*p* < 0.05).

### 3.4. Serum Antioxidant Index and Relevant Gene Expression in the Liver

According to Table 7, T-SOD activity in CON was significantly higher than that in HB and HL groups (*p* < 0.05), and HLA was significantly higher than that in HB group (*p* < 0.05); MDA concentrations in the CON and HLA were lower than those in the HB and HL (*p* < 0.05), and the HLA was higher than the CON (*p* < 0.05). GPx concentration in the CON was higher than in the HB and HL (*p* < 0.05), the HB was lower than all other groups (*p* < 0.05), and the HLA showed no significant difference compared with CON and HL (*p* > 0.05). From Figure 2, the *SOD1* and *GPx* in the liver were up-regulated in CON, HL and HLA compared to HB (*p* < 0.05).

### 3.5. Serum Immunoglobulin, Inflammatory Cytokine Indicators and Relevant Gene Expression in the Liver

As presented in Table 8 and Table 9, the HLA had the highest concentrations of IgA, IgG, and IgM compared to the other groups (*p* < 0.05). Specifically, the CON had higher IgA and IgM concentration compared to the HB and HL (*p* < 0.05), while the HB had lower IgG and IgM concentration compared to the HL (*p* < 0.05). IL-1β concentration was higher in the HB compared to other groups (*p* < 0.05), while the HL and HLA were not significantly different from each other (*p* > 0.05), but were both higher than CON (*p* < 0.05). For IL-10, the HLA showed a higher concentration than the other groups (*p* < 0.05), which did not differ significantly from each other (*p* > 0.05). As shown in Figure 3, in the liver, the *IL-1β* mRNA level was highest in HB (*p* < 0.01), and was also higher in HL and HLA than in CON (*p* < 0.05). The relative expression level of *IL-10* mRNA in HLA was significantly higher than that in other groups (*p* < 0.05).

### 3.6. Serum Biochemical Indicators and Relevant Gene Expression in the Liver

According to Table 10, T-CHO concentration was lowest in the CON compared to the other groups (*p* < 0.05). In addition, the HL and HLA had lower T-CHO concentrations and AST activity than the HB (*p* < 0.05). Moreover, the HL exhibited higher concentration of HDL-C than the CON (*p* < 0.05), while the HB showed higher concentration of LDL-C compared to the CON and HLA (*p* < 0.05). Regarding TG and GLU content, the CON had lower concentration than the TRT (*p* < 0.05), while the HL and HLA had lower concentration than the HB (*p* < 0.05). From Figure 4, in the liver, the HMGCR mRNA level was highest in HB (*p* < 0.01), and was also higher in HL and HLA than in CON (*p* < 0.05). LDLR was up-regulated in CON, HL and HLA compared to HB (*p* < 0.05), and also higher in HL and HLA than in CON (*p* < 0.05).

## 4. Discussion

### 4.1. Production Performance, Follicle Development, Serum Hormone and Related Gene Expression in the Ovary

The results showed that heat stress led to a severe decline in EPR, and that low-protein amino acid-balanced diets and APSs offset some of the effects of heat stress, but still not to the same extent as the CON. Furthermore, the trend in FCR was consistent with EPR. Previous studies have shown that high environmental temperatures disadvantageously influence the production performance of birds, probably due to long-term behavioral, metabolic and physiological changes in response to heat stress [21,22]. A lot of research has shown that high dietary CP can result in reduced protein use and increased heat output, while restricting consumption of protein and adding synthetic DL-methionine and L-lysine could increase the efficiency of hens’ protein use and productivity [13,23]. CP has a higher thermal gain and is conducive to more metabolism heat production compared to fat and carbohydrate [24]. Therefore, a low-protein diet effectively mitigates heat stress and boosts protein efficiency in laying hens. This process is achieved by supplementing amino acids (AAs) in order to reduce feeding cost [25,26] and environmental pollution [27]. The mechanism of increased FCR in heat-stressed laying hens fed a low-CP diet is not clear. Reduction in dietary CP (1–3%) did not significantly affect the egg weight and egg production performance of laying hens with a balanced amino acid diet [28,29]. Moreover, it has been reported that if synthetic AA (methionine, lysine, threonine, tryptophan and ILE) is added, the CP in the egg diets can be reduced by 2% [30].

APS has many beneficial properties, such as anti-oxidation [31], immune regulation [32,33], anti-inflammatory [34], hypoglycemic [35] and other biological activities. For increased reactive oxygen species in laying hens due to heat stress, APS can be a kind of inhibiting agent to decrease the side effects of heat stress. Normal ovulation relies on the development of hierarchical follicles [36]. This study showed that the number of SWF in the HL and HLA groups was higher than in the HB group, which may be due to the APSs, which can promote blood circulation of laying hens and further promote follicular development. The LPAB diets can reduce the adverse effects of heat stress, and APSs also have antioxidant and anti-inflammatory effects, which have positive effects on protecting follicular health.

ACTH is the main regulator of adrenal nutrition and fascicular steroidogenesis [37]. Under the conditions of this experiment, the high-temperature environment significantly reduced the ACTH concentration in the serum of laying hens. In birds, gonadotropin-releasing hormone released by the hypothalamus stimulates pituitary secretion of FSH and LH, supporting follicular development and ovulation [38]. FSH and LH can promote the formation of oocytes by stimulating the production of ovarian gonadal hormones (such as E_2_, P_4_, etc.) [39]. Studies have shown that the release of ACTH and CORT in serum is affected under heat stress [40]. Studies have found that elevated serum ACTH and CORT concentration under heat stress conditions inhibit the secretion of FSH and LH [41]. In this experiment, the addition of APSs in the LPAB diet could reduce the concentration of ACTH and CORT in serum, which was basically consistent with the above results. It can be inferred that the possible mechanism is that the protein is metabolized in the body to provide precursor products for hormone synthesis, or it may be directly involved in the normal regulation of the hypothalamus–pituitary–ovarian axis that affects the secretion of gonadotropin, thereby reducing the synthesis, storage and secretion of FSH and LH, while APSs mainly regulate the secretion of gonadotropin by reducing the concentration of CORT in serum, thereby increasing the concentration of FSH and FSHR mRNA expression to regulate primary follicle development and increase egg production rate.

### 4.2. Effects of Astragalus Polysaccharides in Low-Protein Diet on Egg Quality

Research has shown that most egg quality traits remain unaffected by protein level [42]. Additionally, observations have indicated that eggshell characteristics do not change when the methionine content in the feed is increased [43]. Haugh unit scores were significantly lower in hens fed 13% CP diet than 16% CP diet, but yolk color was significantly higher than in hens fed 16% CP diet [23]. Low-protein diets contain a high proportion of maize, which is rich in lutein; adding more maize to the diet can improve yolk pigmentation [44]. Research indicates that appropriate levels of EAAs can prevent the harmful effects of low-CP diets on shell quality [45]. Eggshell strength was also not influenced by dietary CP levels, suggesting that the addition of L-threonine and lysine to low-crude-protein diets can maintain eggshell quality when digestible sulfur-containing amino acid requirements are considered.

### 4.3. Serum Antioxidant and Relevant Gene Expression in the Liver

The SOD activity and MDA content are important indicators of antioxidant performance. GPx is an important peroxidase enzyme widely present in the body, and is one of the indicators of the body’s resistance to peroxidation. It has been shown that high-temperature conditions induce the production of free radicals and oxidative stress, which induce lipid peroxidation and cell membrane damage [46]. A decrease in SOD activity and GPx concentration, consistent with an increase in free radical production in the high-temperature group, has been confirmed [47]. Meanwhile, findings indicate that heat stress increases the levels of SOD and MDA in broiler livers [48]. Nrf-2, a key transcription factor in the antioxidant system, protects cells by upregulating antioxidants like SOD and GPx to counteract stressors [49], which is consistent with the findings of this study. Low-protein diets, supplemented with essential amino acids to meet animals’ nutritional needs and maintain amino acid balance, can boost antioxidant enzyme synthesis and enhance antioxidant capacity [50]. Beneficial effects of feeding low-protein diets (with recommended ME) under heat stress have been observed, and this finding is consistent with several other studies conducted under similar conditions [51,52]. Furthermore, it has been shown that oxidative stress dominates the damage caused by heat stress; heat stress causes enhanced metabolism in livestock organisms, causing excessive accumulation of ROS and causing an oxidative–antioxidant imbalance in the organism and oxidative stress; this in turn causes oxidative damage to tissue cells, proteins and nucleic acids [53,54].

### 4.4. Serum Immunoglobulin, Inflammatory Cytokines and Relevant Gene Expression in the Liver

IgA, IgG and IgM play an important role in the immune process of poultry [55]. IgM is an early antibody of humoral immunity, and IgG has the function of neutralizing viruses and bacteria [56]. Supplementation of 0.498% tryptophan in low-protein diet (14.05% CP) significantly increased serum IgG and IgM contents in broilers [57]. Studies have reported that heat stress can inhibit the production of chicken antibodies, and the production of IgA is the lowest [58], which is consistent with the results of [59]. In this experiment, high temperature reduced the content of IgA, IgG and IgM in serum, and low-protein diet significantly increased the content of IgG and IgM, and the low-protein diet supplemented with APSs reversed the adverse effects of high temperature and increased the content of immunoglobulin.

IL-1β is mainly released by monocytes, macrophages and non-immune cells during cell injury and infection, which can promote the production of other cytokines and play an important role in immune response [60]. IL-10 exerts strong anti-inflammatory effects by suppressing pro-inflammatory cytokine production and modulating cytokine receptor expression [61]. Under the conditions of this experiment, the high-temperature environment increased the content of IL-1β in the serum of laying hens. The concentration of IL-1β in the serum of LPAB diets decreased by 14.4% and 17.00%, respectively, and IL-1β relative mRNA expression in the liver was decreased, but it still failed to completely offset the damage caused by heat stress. After adding APSs to the LPAB diet, the concentration of IL-10 increased by 38.67%, and IL-10 relative mRNA expression in the liver was upregulated, indicating that although the high temperature destroyed the balance of its cytokines and adversely affected the immune system of laying hens, the LPAB diets and APSs started the anti-inflammatory reaction. At the same time, it also inhibits inflammation, and reduces body damage.

### 4.5. Serum Biochemical and Relevant Gene Expression in the Liver

Cholesterol and triglyceride are mainly related to lipid metabolism in laying hens. The contents of CHOL and LDL-C in 21-day-old male broilers were significantly increased under heat stress [62]. APSs could reverse the increase in serum T-CHO, TG and LDL-C levels and the decrease in serum HDL-C levels induced by alcohol in mice [63]. Under the condition of high-temperature stress, the addition of APSs to the diet can significantly reduce the content of total cholesterol and LDL-C in serum, which is basically consistent with the above studies. It may be due to the fact that APSs are good lipid regulators and can improve plasma lipoprotein levels [59,64]. Under the condition of this experiment, heat stress and low-protein diet led to the increase in total cholesterol and triglyceride content in serum of laying hens, which may be due to the decrease in basal metabolism of laying hens in high-temperature environment or low-protein state, affecting the normal metabolism of lipids in vivo, and the decrease in lipid concentration. The energy required for cell metabolism is only provided by related enzymes to decompose nutrients, so the contents of TG and T-CHO involved in lipid metabolism are significantly increased. ALT and AST are liver function markers associated with amino acid metabolism, and their elevated levels under stress reflect tissue damage [65,66]. In this experiment, the addition of APSs in the diet can reduce the activity of AST in serum, protect the liver and reduce the damage caused by heat stress. In different treatments of low-crude-protein diets, no significant changes were observed in TP, ALB, GLB, and AST (standard CP 2, 4, and 6% control diets) [67]. It is speculated that the addition of APSs to the low-protein diet has no obvious effect on improving the protein metabolism of laying hens under high-temperature environment. In addition, heat stress can destroy the integrity of the intestinal barrier, promote the absorption of glucose, and increase serum glucose content [68]. Giving 4% low-protein diet to pregnant mice will lead to dysfunction of islet β cells in their offspring and reduce the secretion of insulin and amylin [69]. Studies have shown that APSs can reduce serum GLU levels, increase insulin sensitivity, and reduce insulin resistance [70]. The results of this experiment are basically consistent with the above results. Some studies found that when broilers were fed low-protein diets (2–4% CP), there was no change in serum total protein [71,72]. Further observations indicated that total protein is only affected when diets lack amino acids [73]. Thus, meeting AA needs seems more critical than CP itself. As shown in this study, low-CP diets can boost hepatic lipogenesis, raising triglyceride levels. Cholesterol biosynthesis starts with acetyl-CoA, with *HMGCR* acting as the rate-limiting enzyme [74]. Cells also take up cholesterol via *LDLR*-mediated endocytosis [75]. Our data showed significantly upregulated *HMGCR* expression in HB livers. Compared to HB, hepatic *LDLR* expression increased by 53% in HL and 45% in HLA, suggesting heat stress boosts blood cholesterol by reducing extrahepatic cholesterol absorption. The LPAB diets and APSs can both enhance cholesterol uptake and suppress its synthesis.

## 5. Conclusions

In conclusion, heat stress caused by a high-temperature environment reduces production performance. LPAB diets and APSs can reduce the adverse effects of heat stress, improving FCR and follicular development, increasing egg yolk color, and adding 0.5% APSs can also enhance EPR. In addition, LPAB diets and APSs significantly improve antioxidant and immune performance in heat stress. Based on LPAB diets, this study added APSs to explore its effect on laying performance, antioxidant performance, immune performance and lipid metabolism of laying hens. This provides a reference for the use of LPAB diets and plant-derived feed additives in laying hen production. At the same time, it provides a feasible method to mitigate the negative impact of high-temperature environments on laying hen production. APSs can improve the health status of laying hens, and it is expected to be a green, healthy, production-friendly, and antioxidant feed additive.

## Figures and Tables

**Figure 1 animals-15-02385-f001:**
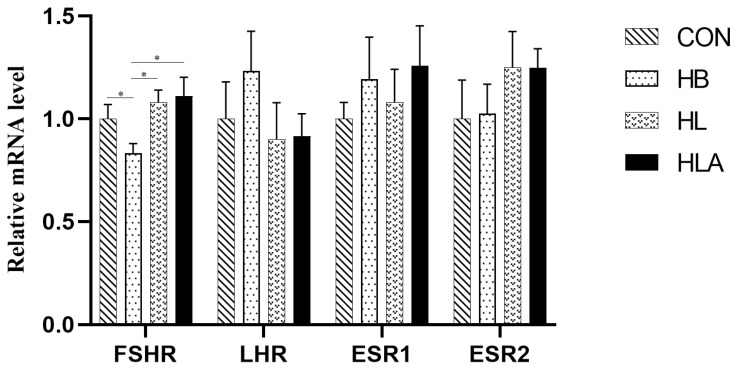
Relative gene expression of productive hormone-related genes. CON: a group was maintained at 24 °C with a basal diet; HB: a group was exposed to 32 °C, fed basal diet; HL: a group was exposed to 32 °C, low-protein amino acid-balanced diet; HLA: a group was exposed to 32 °C low- protein amino acid-balanced diet with 0.5% *Astragalus* polysaccharides. *n* = 6; * indicates *p* < 0.05 relative to the control group. All data are expressed as mean ± SEM in the figure.

**Figure 2 animals-15-02385-f002:**
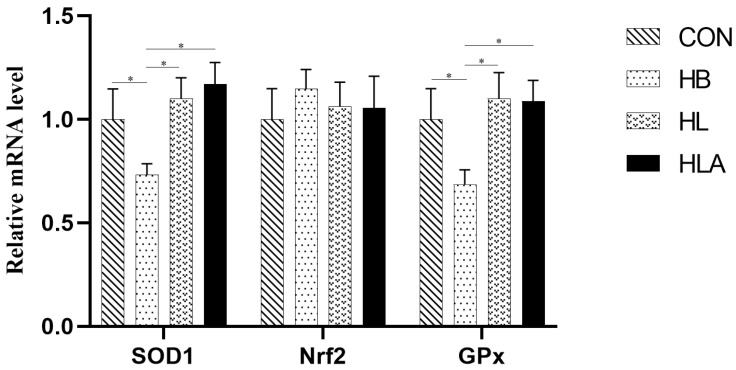
The mRNA expression of antioxidant-related genes. CON: a group was maintained at 24 °C with a basal diet; HB: a group was exposed to 32 °C, fed basal diet; HL: a group was exposed to 32 °C, low-protein amino acid-balanced diet; HLA: a group was exposed to 32 °C low-protein amino acid-balanced diet with 0.5% *Astragalus* polysaccharides. *n* = 6; * indicates *p* < 0.05 relative to the control group. All data are expressed as mean ± SEM in the figure.

**Figure 3 animals-15-02385-f003:**
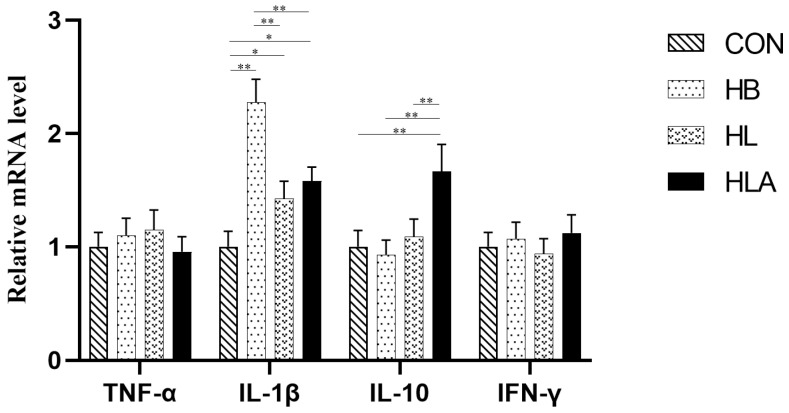
The mRNA expression of inflammatory-related genes. CON: a group was maintained at 24 °C with a basal diet; HB: a group was exposed to 32 °C, fed basal diet; HL: a group was exposed to 32 °C, low-protein amino acid-balanced diet; HLA: a group was exposed to 32 °C low-protein amino acid-balanced diet with 0.5% *Astragalus* polysaccharides. *n* = 6; * indicates *p* < 0.05 and ** indicates *p* < 0.01 relative to the control group. All data are expressed as mean ± SEM in the figure.

**Figure 4 animals-15-02385-f004:**
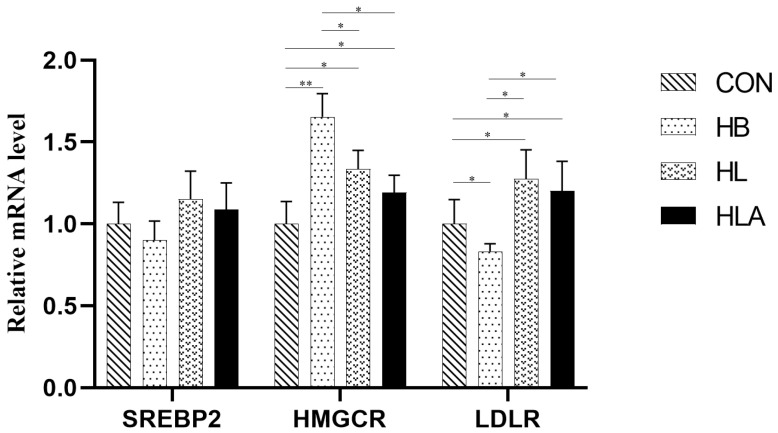
The mRNA expression of reproductive lipid metabolism-related genes. CON: a group was maintained at 24 °C with a basal diet; HB: a group was exposed to 32 °C, fed basal diet; HL: a group was exposed to 32 °C, low-protein amino acid-balanced diet; HLA: a group was exposed to 32 °C low-protein amino acid-balanced diet with 0.5% *Astragalus* polysaccharides. *n* = 6; * indicates *p* < 0.05 and ** indicates *p* < 0.01 relative to the control group. All data are expressed as mean ± SEM in the figure.

**Table 1 animals-15-02385-t001:** Composition and nutrient levels of basal diets (g/kg air-dry basis).

Ingredients	Basic Diet	Low-Protein Diet
Corn	632	672
Soybean meal	230	163
Wheat bran	33.5	57.5
Calcium dihydrogen phosphate	6.30	6.30
Limestone	91.2	91.7
Sodium chloride	3.00	3.00
DL-Methionine (990 g/kg)	1.50	1.80
L-lysine HCl	0.00	1.60
L-threonine	0.00	0.90
L-tryptophan	0.00	0.40
^1^ Premix	2.50	2.50
^2^ Nutrient components		
ME (MJ/kg)	11.0	11.0
Crude protein	154	132
Crude fiber	25.9	24.3
Calcium	35.5	35.5
Total phosphorus	5.00	4.90
Nonphytate phosphorus	2.42	2.36
Methionine	3.86	3.88
Lysine	7.81	7.81
Threonine	5.85	5.91
Tryptophan	1.81	1.83

^1^ The premix provided following per kg of the diet: VA 225 K IU, VD_3_ 60 K IU, VE 360 K IU, VK_3_ 35 mg, VB_12_ 0.30 mg, Nicotinic 700 mg, Cu (CuSO_4_·5H_2_O) 500 mg, Fe (FeSO_4_·H_2_O) 990 mg, Zn (ZnSO_4_·H_2_O) 1130 mg, Mn (MnSO_4_·H_2_O) 1700 mg. ^2^ The contents of methionine, lysine, threonine and tryptophan were measured values and others were calculated values.

**Table 2 animals-15-02385-t002:** Sequences of the primers for the target genes.

Target Genes	Primer Sequences (5′−3′)	Size/bp	Accession No.
*FSHR*	F:ACATTCCCACCAATGCCACA R:AGTGCACCTTATGGACGACG	300	NM_205079.1
*LHR*	F:GGGCTTTCCCAAGCCTACAT R:TGGTGTCTTTATTGGCGGCT	133	NM_204936.1
*ESR1*	F:GCTCTCACCCTTCATCCAT R:GACATCCTCTCACGAATGC	150	NC 006090.3
*ESR2*	F:AGAGAACGCTGTGGGTAT R:TAGGACGACTCACCAACA	187	NC 006092.3
*β-actin*	F:TGGATGATGATATTGCTGC R:ATCTTCTCCATATCATCCC	253	K02173
*SOD1*	F:TTGTCTGATGGAGATCATGGCTTC R:TGCTTGCCTTCAGGATTAAAGTGAG	98	NM_205064.1
*Nrf2*	F:GGGACGGTGACACAGGAACAAC R:TCCACAGCGGGAAATCAGAAAGATC	93	NM_205117.1
*GPx*	F:ACGGCGCATCTTCCAAAG R:TGTTCCCCCAACCATTTCTC	288	NM_001277853
*TNF-α*	F:GCCCAGTTCAGATGAGTTGCC R:AAGAGGCCACCACACGACAG	100	NC_052545.1
*IL-1β*	F:TGCCTGCAGAAGAAGCCTCG R:GACGGGCTCAAAAACCTCCT	204	NC_052553.1
*IL-10*	F-CAGACCAGCACCAGTCATCA R-TCCCGTTCTCATCCATCTTCTC	163	NM_001004414.2
*IFN-γ*	F:TGATGGCGTGAAGAAGGTG R:GACTGGCTCCTTTTCCTTTTG	150	AJ012245
*SREBP2*	F:TGGGCGACATAGACGAGATG R:CACCGCCACCCTGGAAG	102	XM 015289037
*HMGCR*	F:TTGGATAGAGGGAAGAGGGAAG R:TTGGATAGAGGGAAGAGGGAAG	137	NM 204485.3
*LDLR*	F:CATCAGCTTCGGGAACCCTC R:CTGTGCACACTCCGCTGT	96	NM 204452.1

**Table 3 animals-15-02385-t003:** Effects of *Astragalus* polysaccharides in low-protein diet on production performance of laying hens.

Items ^1^	CON	HB	HL	HLA	SEM ^2^	*p*-Value
EPR (%)	83.30 ^a^	73.27 ^c^	76.22 ^bc^	78.79 ^ab^	0.992	<0.001
Breakage rate (%)	0.17	0.34	0.36	0.24	0.040	0.318
Mortality (%)	0	0.15	0.11	0.14	0.046	0.674
ADFI (g)	103.14	106.04	101.70	100.50	0.775	0.059
Average egg weight (g)	64.67	63.22	63.69	63.45	0.848	0.364
FCR (g/g)	1.95 ^b^	2.30 ^a^	2.11 ^b^	2.04 ^b^	0.035	<0.001

^a–c^ Mean values with different superscripts in the same row indicate significant differences between treatments (*p* < 0.05). *n* = 8. ^1^ EPR, egg production rate; FCR, feed conversion ratio; ADFI, average daily feed consumption. ^2^ SEM: standard error of mean.

**Table 4 animals-15-02385-t004:** Effects of *Astragalus* polysaccharides in low-protein diet on follicular development of laying hens.

Items ^1^	CON	HB	HL	HLA	SEM ^2^	*p*-Value
SWF	33.00 ^a^	26.65 ^b^	31.00 ^a^	30.13 ^a^	0.654	0.001
BYF	6.38	6.88	6.88	7.00	0.265	0.857
GF	5.25	5.00	5.00	5.25	0.188	0.118

^a,b^ Mean values with different superscripts in the same row indicate significant differences between treatments (*p* < 0.05). *n* = 6. ^1^ SWF, small white follicles; BYF, big yellow follicle; GF, graded follicle. ^2^ SEM: standard error of mean.

**Table 5 animals-15-02385-t005:** Effects of *Astragalus* polysaccharides in low-protein diet on egg quality.

Items ^1^	CON	HB	HL	HLA	SEM ^2^	*p*-Value
Albumen height (mm)	6.66	6.80	6.68	6.68	0.129	0.312
Haugh unit	78.91	82.40	78.25	84.20	1.051	0.138
Yolk color	5.17 ^c^	5.85 ^bc^	6.38 ^ab^	6.46 ^a^	0.131	<0.001
Yolk ratio (%)	26.29	25.48	25.43	25.39	0.002	0.115
Eggshell strength (N)	4.17	4.05	4.11	4.19	0.110	0.971
Shell thickness (mm)	0.29	0.29	0.29	0.27	0.005	0.324
Egg shape index	1.31	1.29	1.32	1.30	0.005	0.507

^a–c^ Mean values with different superscripts in the same row indicate significant differences between treatments (*p* < 0.05). *n* = 24. ^1^ The control group was maintained at 24 °C with a basal diet (CON), while all experimental groups were exposed to 32 °C, fed a basal diet (HB), a low-protein amino acid-balanced (LPAB) diet (HL), LPAB diet with 0.5% *Astragalus* polysaccharides (HLA). ^2^ SEM: standard error of mean.

**Table 6 animals-15-02385-t006:** Effects of *Astragalus* polysaccharides on serum hormone indexes in low-protein diet under high-temperature environment.

Items ^1^	CON	HB	HL	HLA	SEM ^2^	*p*-Value
ACTH (pg/mL)	40.0 ^b^	68.3 ^a^	39.4 ^b^	43.0 ^b^	2.36	<0.001
CORT (ng/mL)	69.5 ^c^	86.5 ^a^	80.3 ^b^	73.9 ^c^	1.47	<0.001
E_2_ (pg/mL)	376	373	366	381	3.45	0.195
FSH (mIU/mL)	12.25 ^a^	8.39 ^b^	11.70 ^a^	12.53 ^a^	0.49	0.005
LH (mIU/mL)	14.20	13.71	13.38	13.57	0.56	0.055
P_4_ (pmol/mL)	1.20	1.23	1.20	1.27	0.03	0.151

^a–c^ Mean values with different superscripts in the same row indicate significant differences between treatments (*p* < 0.05). *n* = 6. ^1^ The control group was maintained at 24 °C with a basal diet (CON), while all experimental groups were exposed to 32 °C, fed a basal diet (HB), a low-protein amino acid-balanced (LPAB) diet (HL), LPAB diet with 0.5% *Astragalus* polysaccharides (HLA). CORT, corticosterone; E_2_, estradiol; FSH, follicle-stimulating hormone; LH, luteinizing hormone; P_4_, progesterone. ^2^ SEM: standard error of mean.

**Table 7 animals-15-02385-t007:** Effects of *Astragalus* polysaccharides in low-protein diet on antioxidation performance of serum.

Items ^1^	CON	HB	HL	HLA	SEM ^2^	*p*-Value
T-SOD (U/mL)	143.66 ^a^	126.51 ^c^	133.49 ^b^	137.21 ^ab^	1.593	<0.001
MDA (nmol/mL)	234.24 ^c^	338.78 ^a^	329.64 ^a^	289.27 ^b^	8.231	<0.001
GPx (U/mL)	95.53 ^a^	85.48 ^c^	88.89 ^b^	91.93 ^ab^	0.973	<0.001
T-AOC (U/mL)	6.28	5.76	6.41	6.41	0.116	0.145

^a–c^ Mean values with different superscripts in the same row indicate significant differences between treatments (*p* < 0.05). *n* = 6. ^1^ The control group was maintained at 24 °C with a basal diet (CON), while all experimental groups were exposed to 32 °C, fed a basal diet (HB), a low-protein amino acid-balanced (LPAB) diet (HL), LPAB diet with 0.5% *Astragalus* polysaccharides (HLA). T-SOD, total superoxide dismutase; MDA, malondialdehyde; GPx, glutathione peroxidase; T-AOC, total antioxidant capacity. ^2^ SEM: standard error of mean.

**Table 8 animals-15-02385-t008:** Effects of *Astragalus* polysaccharides on serum immunoglobulin content of laying hens under high-temperature environment in low-protein diet.

Items ^1^	CON	HB	HL	HLA	SEM ^2^	*p*-Value
IgA (μg/mL)	157.83 ^b^	139.11 ^c^	144.97 ^c^	256.07 ^a^	8.69	<0.001
IgG (mg/mL)	2.14 ^b^	2.01 ^c^	2.11 ^b^	2.62 ^a^	0.05	<0.001
IgM (μg/mL)	651.80 ^b^	482.49 ^d^	582.76 ^c^	671.91 ^a^	13.55	<0.001

^a–d^ Mean values with different superscripts in the same row indicate significant differences between treatments (*p* < 0.05). *n* = 6. ^1^ The control group was maintained at 24 °C with a basal diet (CON), while all experimental groups were exposed to 32 °C, fed a basal diet (HB), a low-protein amino acid-balanced (LPAB) diet (HL), LPAB diet with 0.5% *Astragalus* polysaccharides (HLA). IgA, immunoglobulin A; IgG, immunoglobulin G; IgM, immunoglobulin M. ^2^ SEM: standard error of mean.

**Table 9 animals-15-02385-t009:** Effects of *Astragalus* polysaccharides on serum inflammatory cytokines in laying hens under high-temperature environment in low-protein diet.

Items ^1^	CON	HB	HL	HLA	SEM ^2^	*p*-Value
IL-1β (pg/mL)	473.35 ^c^	598.33 ^a^	511.92 ^b^	496.64 ^b^	8.89	<0.001
IL-2 (pg/mL)	236.64	254.43	256.30	232.06	5.18	0.259
IL-6 (pg/mL)	21.88	26.62	24.61	21.00	0.86	0.074
IL-10 (pg/mL)	48.46 ^b^	45.60 ^b^	44.45 ^b^	61.64 ^a^	1.40	<0.001
MCP-1 (ng/mL)	0.58	0.50	0.59	0.53	0.02	0.152
TNF-α (pg/mL)	45.89	53.03	53.35	48.21	6.34	0.127
IFN-γ (pg/mL)	66.00	63.88	69.09	70.84	8.32	0.253

^a–c^ Mean values with different superscripts in the same row indicate significant differences between treatments (*p* < 0.05). *n* = 6. ^1^ The control group was maintained at 24 °C with a basal diet (CON), while all experimental groups were exposed to 32 °C, fed a basal diet (HB), a low-protein amino acid-balanced (LPAB) diet (HL), LPAB diet with 0.5% *Astragalus* polysaccharides (HLA). IL-1β, interleukin 1 beta; IL-2, interleukin 2; IL-6, interleukin 6; IL-10, interleukin 10; MCP-1, monocyte chemokine-1; TNF-α, tumor necrosis factor-α; IFN-γ, interferon-γ. ^2^ SEM: standard error of mean.

**Table 10 animals-15-02385-t010:** Effects of *Astragalus* polysaccharides on serum biochemical indexes under high-temperature environment in low-protein diet.

Items ^1^	CON	HB	HL	HLA	SEM ^2^	*p*-Value
T-CHO (mmol/L)	2.51 ^c^	3.91 ^a^	3.28 ^b^	3.00 ^b^	0.29	<0.001
HDL-C (mmol/L)	0.54 ^b^	0.60 ^ab^	0.63 ^a^	0.59 ^ab^	0.01	0.041
LDL-C (mmol/L)	1.26 ^b^	1.42 ^a^	1.33 ^ab^	1.26 ^b^	0.02	0.019
TG (mmol/L)	12.63 ^c^	20.33 ^a^	16.41 ^b^	17.96 ^b^	0.60	<0.001
AST/(U/L)	24.17 ^a^	24.83 ^a^	18.83 ^b^	20.67 ^b^	0.72	0.004
ALT (U/L)	180.35	186.82	185.01	174.02	5.98	0.313
TP (g/L)	52.50	55.68	54.28	56.17	0.72	0.279
ALB (g/L)	14.39	15.33	15.30	14.99	0.25	0.522
GLB (g/L)	38.00	38.00	39.00	39.33	0.45	0.657
A/G	0.38	0.41	0.39	0.38	0.01	0.791
GLU (mmol/L)	6.04 ^d^	7.17 ^c^	7.95 ^a^	7.67 ^b^	0.14	<0.001

^a–d^ Mean values with different superscripts in the same row indicate significant differences between treatments (*p* < 0.05). *n* = 6. ^1^ The control group was maintained at 24 °C with a basal diet (CON), while all experimental groups were exposed to 32 °C, fed a basal diet (HB), a low-protein amino acid-balanced (LPAB) diet (HL), LPAB diet with 0.5% *Astragalus* polysaccharides (HLA). T-CHO, total cholesterol; HDL-C, high-density lipoprotein cholesterol; LDL-C, low-density lipoprotein cholesterol; TG, triglyceride; TP, total protein; ALB, albumin; GLB, globulin; ALT, alanine aminotransferase; AST, aspartate aminotransferase; A/G, albumin/globulin; GLU, glucose. ^2^ SEM: standard error of mean.

## Data Availability

The original contributions presented in this study are included in the article. Further inquiries can be directed to the corresponding author.

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
