# Peer review of "Effects of Low-Protein Amino Acid-Balanced Diets and Astragalus Polysaccharides on Production Performance, Antioxidants, Immunity, and Lipid Metabolism in Heat-Stressed Laying Hens"

_animals, 2025, doi:10.3390/ani15162385_

Round 1

Reviewer 1 Report

Comments and Suggestions for Authors

Despite the significance of this report, there are some issues that should be addressed:

  1. LN 98: What is the NRC (2012)? Do you have this reference? If so, please share with us and cite in the bibliography section.
  2. Based on the bird's age (52 wk), the birds was classified in the third period of production and the egg production should range from 88 to 91% according to standards. What feed intake level did you assume when formulating the basal diet?

Author Response

Comments 1: LN 98: What is the NRC (2012)? Do you have this reference? If so, please share with us and cite in the bibliography section.

Response 1: Thank you very much for catching the citation error. After checking the original source, the correct reference is indeed: National Research Council (NRC). (1994). Nutrient Requirements of Poultry (9th rev. ed.). Washington, DC: National Academy Press. I have revised the text and reference list accordingly, replacing every instance of “NRC (2012)” with “NRC (1994)” and adding the full bibliographic details above. I appreciate your careful attention, which has helped improve the accuracy of the manuscript. Page 3, 2.1. Experimental Design and Diets, and line 88ï¼›Page 17, References, and line 538-539.

Comments 2: Based on the bird's age (52 wk), the birds was classified in the third period of production and the egg production should range from 88 to 91% according to standards. What feed intake level did you assume when formulating the basal diet?

Response 2: Thank you for highlighting the difference between the Hy-Line Brown performance guide and the actual flock performance. At 52 weeks of age, the birds were indeed in the third phase of production, and according to the Hy-Line Brown Management Guide, the expected egg production under optimal conditions ranges from 88 to 91%, with a daily feed intake of approximately 99-105 g/bird/day during the 52-58 week period. In our study, the actual egg production rate (CON) was around 83%, which is slightly below the standard but consistent with findings reported by Chen et al. (2024) under similar conditions. Real-world variation in flock management, which may not fully match the idealized conditions of breeding manuals, and possible genetic or flock-level variation.

In our study, the actual average daily feed intake was approximately 105 g/bird, as determined during a 10-day pre-experimental adaptation period. To account for potential feed losses due to spillage and wastage, we slightly increased the feed allocation to 110 g/bird/day when formulating the basal diet, ensuring that nutrient intake would not be compromised. We chose to formulate the diets based on actual performance and intake data to better reflect practical conditions and improve the applicability of our findings under field-relevant scenarios.

Cite: Yaping C, Yunwei H, Yuan Z, et al. Effects of conjugated linoleic acid on production performance, biochemical indexes and antioxidant properties of laying hens in late laying period[J]. Journal of Northwest A & F University-Natural Science Edition, 2024, 52(10).

Reviewer 2 Report

Comments and Suggestions for Authors

Climate change, which significantly contributes to the occurrence of heat stress, remains a widely explored topic in the scientific literature. Nevertheless, further research is still needed to identify effective strategies for mitigating its associated negative effects. In this context, I acknowledge the value of the present study; however, I believe there are several aspects that could be improved, which I will outline below.

-A revision of the keywords is recommended, as they are already included in the title of the paper;

-Why was a base formulation used that does not comply with the specific requirements of the Hy Line Brown hybrid and relies solely on the NRC standards from the year 2012, which are now considered outdated?

-The experimental period should be specified in Chapter Material and Methods in order to ensure methodological transparency and reproducibility of the study;

-This chapter should also include detailed information on the additive used, such as the method of extraction or production, form of administration, method of inclusion in the formulation, and other relevant characteristics essential for understanding the experimental protocol;

-What test was used to assess the homogeneity of variance within the experimental groups, and, if this assumption was not met, which statistical methods were applied to compare groups with unequal variances? Assessing the homogeneity of variance within experimental groups represented a critical preliminary step prior to conducting comparative analyses. Failure to meet this assumption may compromise the accuracy of statistical interpretations—potentially leading to Type I or Type II errors—and requires the application of alternative analytical methods capable of handling unequal variance distributions.

-The use of the General Linear Model (GLM) is recommended, as it allows for the simultaneous analysis of multiple factors and their interactions (temperature, protein level, and supplementation with APS). Unlike classical ANOVA, which is limited to evaluating the effect of a single factor, GLM provides a broader analytical framework suitable for multifactorial experiments, thereby enabling a more rigorous and comprehensive interpretation of the data.

-The bibliography is excessively extensive for an original research article and resembles more the structure of a review paper. Moreover, a significant portion of the cited sources is outdated and does not adequately reflect recent developments in the field. Only approximately 18% of the references were published within the last five years, which compromises the currency and relevance of the literature supporting the study. A thorough revision and update of the bibliography is recommended, with an emphasis on including recent, high-quality, and indexed scientific sources.

Author Response

Comments 1: A revision of the keywords is recommended, as they are already included in the title of the paper.

Response 1: Thank you for your helpful suggestion. We agree that the original keywords overlapped with the title. Accordingly, we have revised the keywords to include more specific and complementary terms that reflect the core aspects of the study while avoiding repetition. The updated keywords are as follows:

Keywords: Oxidative stress; Follicular development; Antioxidant enzymes; Immunomodulation; Anti-inflammatory cytokine

The revised version has been updated in the manuscript. Page 2, Keywords, and line 41-42.

Comments 2: Why was a base formulation used that does not comply with the specific requirements of the Hy Line Brown hybrid and relies solely on the NRC standards from the year 2012, which are now considered outdated?

Response 2: Thank you for raising this important point. Firstly, we sincerely apologize for the typographical error in the manuscript, where we mistakenly cited “NRC 2012” instead of the correct “NRC 1994.” Then, we fully recognise that Hy-Line Brown has its own, more precise nutrient specifications and that the NRC (1994) values are now regarded as minimal rather than optimal. Although the 1994 NRC guidelines were used as a baseline reference, we did not rely on them exclusively. In fact, the actual nutrient intake values achieved in our study were closely aligned with the recommendations in the Hy-Line Brown Management Guide (2024) for 52-week-old hens. Based on an average daily feed intake of 105 g/bird (determined during a 10-day pre-experimental period under thermoneutral conditions), each hen received:

Metabolizable Energy: 1.15 MJ/day (vs. 1.26 MJ/day recommended)

Crude Protein: 16.17 g/day(vs. 16.30/day recommended)

Total Amino Acids (per bird per day):

Lysine: 820 mg (vs. 816 mg recommended)

Methionine: 405 mg (vs. 401 mg recommended)

Threonine: 614 mg (vs. 614 mg recommended)

Tryptophan: 190 mg (vs. 191 mg recommended)

These figures indicate that the diets were carefully formulated to ensure nutritional adequacy, particularly in terms of key amino acid intake, consistent with the hens' age and production phase. Therefore, although NRC (1994) served as a formulation reference, the actual dietary design was closely guided by Hy-Line’s hybrid-specific requirements, ensuring relevance and scientific rigor. Although this practice may limit the absolute production performance of Hy-Line brown-shell laying hens, it does not affect the comparability of different treatments.

The revised version has been updated in the manuscript. Page 3, 2.1. Experimental Design and Diets, and line 88ï¼›Page 17, References, and line 538-539.

Comments 3: The experimental period should be specified in Chapter Material and Methods in order to ensure methodological transparency and reproducibility of the study;

Response 3: Thank you for this helpful suggestion. To enhance methodological transparency and reproducibility, the following sentence has been added to the Materials and Methods. “The experiment was conducted when the hens were between 52 and 58 weeks of age.” The revised version has been updated in the manuscript. Page 3, 2.1. Experimental Design and Diets, and line 81-82.

Comments 4: This chapter should also include detailed information on the additive used, such as the method of extraction or production, form of administration, method of inclusion in the formulation, and other relevant characteristics essential for understanding the experimental protocol.

Response 4: Thank you for prompting us to provide a complete description of the additive. The following paragraph has been inserted in Materials and Methods: The test additive was Astragalus polysaccharide (APS) powder, purity 98 % (w/w), supplied by Xi’an Shennong Technology Co., Ltd. (Xi’an, China). The polysaccharide was extracted from dried Astragalus membranaceus roots by combined ethanol–water extraction, followed by deproteinisation, decolouration and lyophilisation, yielding a water-soluble, off-white powder. Immediately before diet preparation, APS was first pre-blended with ten times its weight of corn meal to ensure homogeneity, then incorporated into the basal mash at 0.5 % (w/w) using a horizontal ribbon mixer (5 min mixing time). The revised version has been updated in the manuscript. Page 3, 2.1. Experimental Design and Diets, and line 97-103.

Comments 5: What test was used to assess the homogeneity of variance within the experimental groups, and, if this assumption was not met, which statistical methods were applied to compare groups with unequal variances? Assessing the homogeneity of variance within experimental groups represented a critical preliminary step prior to conducting comparative analyses. Failure to meet this assumption may compromise the accuracy of statistical interpretations—potentially leading to Type I or Type II errors—and requires the application of alternative analytical methods capable of handling unequal variance distributions.

Respond 5: Thank you for this important methodological question. Prior to all parametric tests, equality of variances was assessed with Levene’s F-test. All examined variables met the homoscedasticity assumption (P>0.05). Consequently, standard one-way ANOVA followed by Tukey’s post-hoc test was used for all between-group comparisons.

The revised version has been updated in the manuscript. Page 6, 2.5. Statistical Analysis, lines 169-170.

Comments 6: The use of the General Linear Model (GLM) is recommended, as it allows for the simultaneous analysis of multiple factors and their interactions (temperature, protein level, and supplementation with APS). Unlike classical ANOVA, which is limited to evaluating the effect of a single factor, GLM provides a broader analytical framework suitable for multifactorial experiments, thereby enabling a more rigorous and comprehensive interpretation of the data.

Respond 6: Thank you for your valuable suggestion regarding the use of the General Linear Model (GLM). We fully agree that GLM is a powerful tool for analyzing multifactorial experimental designs and identifying main and interaction effects between independent variables such as temperature, protein level, and APS supplementation.

In the present study, we designed four treatment groups as follows:

CON-24℃: Basal diet at thermoneutral temperature

HB-32℃: Basal diet at heat stress (32℃)

HL-32℃: Low-protein amino-acid-balanced (LPAB) diet at 32℃

HLA-32℃: LPAB diet + 0.5% APS at 32℃

However, the experimental design was not a fully factorial arrangement. Specifically, the LPAB diet and APS supplementation were only applied under heat stress conditions (32℃), while the thermoneutral group (24℃) received only the basal diet. Therefore, certain factor combinations (e.g., LPAB or APS at 24℃) were not included by design, making it inappropriate to model temperature, protein level, and APS as fully orthogonal factors in a GLM framework.

The decision to adopt this design was based on practical and biological considerations. Firstly, our primary objective was to evaluate the potential of LPAB diets and APS in mitigating heat stress effects, which necessitated focusing those interventions under elevated temperature conditions. Secondly, a fully crossed factorial design would have required a substantially larger sample size and housing capacity, which was not feasible within the scope of this study.

Given these constraints, we employed one-way ANOVA to compare the four predefined treatment groups, followed by post hoc multiple comparisons to assess differences among them. This approach aligns with the structure of our design and ensures statistically valid conclusions.

We appreciate your suggestion, which would indeed be highly appropriate for a fully factorial study in future research.

The revised version has been updated in the manuscript. Page 6, 2.5. Statistical Analysis, lines 170-178. 

Comments 7: The bliography is excessively extensive for an original research article and resembles more the structure of a review paper. Moreover, a significant portion of the cited sources is outdated and does not adequately reflect recent developments in the field. Only approximately 18% of the references were published within the last five years, which compromises the currency and relevance of the literature supporting the study. A thorough revision and update of the bibliography is recommended, with an emphasis on including recent, high-quality, and indexed scientific sources.

Respond 7: Thank you for your valuable feedback. In response to your comment, we have thoroughly revised and updated the bibliography. Outdated or less relevant references have been removed, and a significant number of recent, and high-quality scientific sources have been added. As a result, approximately 60% are now from the past five years, and 93% are now from the past ten years, which we believe greatly enhances the currency and relevance of the literature supporting our study.

We hope these revisions have improved the manuscript’s clarity and focus. Please kindly let us know if any further adjustments are recommended.

These changes have been reflected in the revised reference list and throughout the manuscript.

Reviewer 3 Report

Comments and Suggestions for Authors

Abstract: Format the font size correctly.

Line 57: “Heat stress has been shown to result in a 16.4% 57 decline in feed intake, a 32.6% decrease in body weight, and a 25.6% improvement in the 58 FCR in animals”. "Is this for broilers or laying hens? I recommend revising the introduction to better highlight hens.

Line 57: Correct “egg mortality”.

Line 66: Apoptosis is a process of programmed cell death, meaning it refers to the death of cells, not molecules.

The introduction contains repeated information, particularly regarding heat stress and its effects on ROS formation, feed intake, and related aspects. I suggest summarizing it, especially the first paragraph.

Line 69: “However, reducing the temperature of poultry houses is expensive and ineffective”. I agree that it's expensive, but not ineffective.

Line 70: “The heat gain of crude protein in the diet is higher than that of starch and fat.” I suggest improving this. Crude protein has a higher thermic effect because it generates a greater caloric increment.

Line 95: "I think it's important to explain more clearly how heat stress was implemented. Did you use climate-controlled rooms? In line 107, you mention “Mechanical ventilation combined with wet curtain to control temperature.” How did you maintain 32±1°C for 7 h per day?

Line 98: “formulated to supply the nutrients requirement of the birds (NRC, 2012)”. Does NRC (2012) include requirements for birds?

Line 102: I suggest explaining in more detail how Astragalus Polysaccharides was used. Is it a plant extract or a commercial product?

Line 133: Were all ovaries placed in liquid nitrogen for analysis? Please explain in more detail.

Line 134 – 139: Format the font size correctly.

Line 148: Format the font size correctly.

Line 174: HLB?

Line 175: The FCR of HB worsened significantly, rather than improved.

Table 3: FCR (g/g) or (feed-egg ratio). See line 117.

Table 3: n = 6 or n = 8?

Line 185: Format the font size correctly.

Line 188 – 191: Please standardize the treatment descriptions in the Results section. I suggest using the format from the first paragraph of the Results.

Line 189: Attention, “adding APS can further deepen yolk color (P<0.05).” The Yolk color in HL and HLA was the same.

Line 201: Review “HL had higher concentration of CORT compared to the HB and HLA (P<0.05).”

Line 222: This is confusing “the highest MDA concentration in HB was 338.78 nmol/ml (P<0.05)”

Line 225: Review “and HLA and HB were significantly higher than that in HB 225 (P<0.05).”

Line 248 – 249: Don't repeat values already presented in the tables.

Author Response

Comments 1: Abstract: Format the font size correctly.

Respond 1: Thank you for pointing this out. The font size of the Abstract has been corrected. Page 1,Abstract, lines 30-37.

Comments 2: Line 57: “Heat stress has been shown to result in a 16.4% 57 decline in feed intake, a 32.6% decrease in body weight, and a 25.6% improvement in the 58 FCR in animals”. "Is this for broilers or laying hens? I recommend revising the introduction to better highlight hens.

Respond 2: Thank you for your suggestion. We have revised the sentence in line with your feedback and replaced the reference with a more hen-specific one (Reference 5). Revised text: “(Ezzat et al., 2024).” The revised version has been updated in the manuscript. Page 2, 1.Introduction, lines 49-51; Page 16, References, line 500-503.

Comments 3: Line 57: Correct “egg mortality”.

Respond 3 : We have replaced “egg mortality” with “egg loss” which more accurately conveys the combined losses from reduced egg production, deteriorated shell quality, and increased embryo mortality under heat stress. The revised version has been updated in the manuscript. Page 2, 1.Introduction, lines 51.

Comments 4: Line 66: Apoptosis is a process of programmed cell death, meaning it refers to the death of cells, not molecules.

Respond 4: Thank you for pointing out the wording error. We have revised the sentence:

Revised text:”…heat stress impairs leukocyte protein synthesis, lowers plasma IgG, IgM and anti-inflammatory cytokine concentrations, and increases the levels of pro-inflammatory cytokines.” The revised version has been updated in the manuscript. Page 2, 1.Introduction, lines 53-56.

Comments 5: The introduction contains repeated information, particularly regarding heat stress and its effects on ROS formation, feed intake, and related aspects. I suggest summarizing it, especially the first paragraph.

Respond 5: Thank you for your thoughtful suggestion. In response, we have revised and condensed the first paragraph of the Introduction to remove redundant content, particularly regarding heat stress, ROS formation, and feed intake. Additionally, we have replaced several outdated references with more recent and relevant literature to improve the scientific rigor and clarity of the background section. These changes have been incorporated into the revised manuscript. Page 2, 1.Introduction, lines 45-52, 53-62, 68-70, 68, 70-72.

Comments 6: Line 69: “However, reducing the temperature of poultry houses is expensive and ineffective”. I agree that it's expensive, but not ineffective.

Respond 6: Thank you for this precise comment. We fully agree that active cooling is expensive yet demonstrably effective when properly implemented. To avoid any misinterpretation, the sentence has been revised as follows:”However, these temperature control systems are expensive and hard to maintain consistently in large or open-sided poultry houses.” This change has been made in the revised manuscript. Page 2, 1.Introduction, lines 58-59.

Comments 7: Line 70: “The heat gain of crude protein in the diet is higher than that of starch and fat.” I suggest improving this. Crude protein has a higher thermic effect because it generates a greater caloric increment.

Respond 7: Thank you for your helpful suggestion. We agree that your revised wording provides a more accurate and scientific expression. This change has been made in the revised manuscript. Page 2, 1.Introduction, lines 59-60.

Comments 8: Line 95: "I think it's important to explain more clearly how heat stress was implemented. Did you use climate-controlled rooms? In line 107, you mention “Mechanical ventilation combined with wet curtain to control temperature.” How did you maintain 32±1°C for 7 h per day?

Respond 8: Thank you for highlighting the need for a clearer description of the heat-stress protocol. We used two identical, climate-controlled rooms (12.0×4.5×2.8 m each) equipped with a negative-pressure ventilation system (0.30 m*s-1 air speed). Temperature was automatically regulated every 30 s by a PID controller connected to two 3-kW electric heaters and a variable-speed wet-curtain cooling unit. From 09:00 to 16:00 daily, the set-point was 32 ℃ ; the controller maintained the room at 32±1℃ as recorded by three HOBO data loggers positioned at bird height. Outside the heat-stress window, rooms were kept at 24±1℃ . Relative humidity was simultaneously controlled at 65±5 % via ultrasonic humidifiers.

“Heat stress was applied in climate-controlled rooms. From 09:00 to 16:00 each day, temperature was maintained at 32±1℃ and relative humidity at 65±5% using a PID-regulated combination of electric heaters and a variable-speed wet-curtain system; outside this period the rooms were held at 24±1℃.”This change has been made in the revised manuscript. Page 3, 2.1. Experimental Design and Diets, lines 103-107.

Comments 9: Line 98: “formulated to supply the nutrients requirement of the birds (NRC, 2012)”. Does NRC (2012) include requirements for birds?

Respond 9: Thank you for catching this inaccuracy. The most recent NRC publication for poultry is ”Nutrient Requirements of Poultry,” 9th revised edition (1994); there is no NRC (2012) edition for birds. We revised the manuscript, and the reference list has been updated accordingly. Page 3, 2.1. Experimental Design and Diets, and line 88; Page 17, References, and line 538-539.

Comments 10: Line 102: I suggest explaining in more detail how Astragalus Polysaccharides was used. Is it a plant extract or a commercial product?

Respond 10: Thank you for the suggestion. The following sentence has been added to Materials and Methods: ”The test additive was Astragalus polysaccharide (APS) powder (purity 98 % w/w), a commercial product supplied by Xi’an Shennong Technology Co., Ltd. (Xi’an, China). It was extracted from dried Astragalus membranaceus roots using an ethanol-water combined extraction, followed by deproteinisation, decolourisation and lyophilisation to yield a water-soluble, off-white powder. Immediately before diet preparation, APS was first pre-blended with ten times its weight of corn meal to ensure homogeneity, then incorporated into the basal mash at 0.5 % (w/w) using a horizontal ribbon mixer (5 min mixing time).” The revised version has been updated in the manuscript. Page 3, 2.1. Experimental Design and Diets, and line 97-103.

Comments 11: Line 133: Were all ovaries placed in liquid nitrogen for analysis? Please explain in more detail.

Respond 11: Thank you for this important question. Only ovarian tissues used for gene expression analysis were preserved in liquid nitrogen. Specifically, one hen per replicate was randomly selected. After dissection, the ovary was rinsed with cold PBS, and all visible hierarchical and pre-hierarchical follicles were carefully removed. The remaining ovarian tissue (excluding the follicles) was then rapidly frozen in liquid nitrogen and stored at -80℃ until RNA extraction. The revised version has been updated in the manuscript. Page 4, 2.2. Animals and Sample Collection, and line 130, 133-136.

Comments 12: Line 134 – 139: Format the font size correctly.

Respond 12: Font size for lines 134–139 has been corrected. The revised version has been updated in the manuscript. Page 4, 2.2. Animals and Sample Collection, and line 136-142.

Comments 13: Line 148: Format the font size correctly.

Respond 13: Font size for line 148 has been corrected. The revised version has been updated in the manuscript. Page 4, 2.3. Measurement of Serum Hormone, Antioxidant, Immunoglobulin, Inflammatory and Biochemical Indicators, and line 151.

Comments 14: Line 174: HLB?

Respond 14: Thank you for catching this typo. The instances of “HLB” have been corrected to “HLA” throughout the manuscript. The revised version has been updated in the manuscript. Page 6, 3.1. Production Performance and Follicle Development, and line 184.

Comments 15: Line 175: The FCR of HB worsened significantly, rather than improved.

Respond 15: Thank you for pointing this out. We have corrected line 175 from “increased” to “worsened,” so it now reads: “Compared with CON, HL and HLA, the FCR of HB worsened significantly (P<0.05).” The revised version has been updated in the manuscript. Page 6, 3.1. Production Performance and Follicle Development, and line 185.

Comments 16: Table 3: FCR (g/g) or (feed-egg ratio). See line 117.

Respond 16: Thank you for your suggestion. We agree that the terminology should be consistent throughout the manuscript. The term “feed-egg ratio” appeared only once (in Line 117), and we have now revised it to “FCR” to align with the terminology used in the rest of the manuscript. The revised version has been updated in the manuscript. Page 4, 2.2. Animals and Sample Collection, and line 117.

Comments 17: Table 3: n = 6 or n = 8?

Respond 17: Thank you for spotting this inconsistency. We have corrected Table 3 so that all entries now show n = 8. The revised version has been updated in the manuscript. Page 6, Table 3, and line 190.

Comments 18: Line 185: Format the font size correctly.

Respond: Font size for line 185 has been corrected. The revised version has been updated in the manuscript. Page 6, Table 4, and line 195.

Comments 19: Line 188-191: Please standardize the treatment descriptions in the Results section. I suggest using the format from the first paragraph of the Results.

Respond19: Thank you for this suggestion. We have standardized all treatment descriptions in the Results section to match the format established in the first paragraph. Any deviations from this format have been corrected to maintain uniformity. The revised version has been updated in the manuscript. Page 8, 3.4. Serum Antioxidant Index and Relevant Gene Expression in the Liver, and line 232-236; Page 9, 3.5. Serum Immunoglobulin, Inflammatory Cytokines Indicators and Relevant Gene Expression in the Liver, and line 257-261; Page 11, 3.6. Serum Biochemical Indicators and Relevant Gene Expression in the Liver, and line 287-296.

Comments 20: Line 189: Attention, “adding APS can further deepen yolk color (P<0.05).” The Yolk color in HL and HLA was the same.

Respond 20: Thank you for this careful observation. We have removed the incorrect statement that APS deepened yolk color. The revised version has been updated in the manuscript. Page 6, 3.2. Egg Quality and Egg Components, and line 198-199.

Comments 21: Line 201: Review “HL had higher concentration of CORT compared to the HB and HLA (P<0.05).”

Respond 21: Thank you for this careful reading. We have corrected the sentence to read:

“Additionally, HL had a higher concentration of CORT than HLA (P<0.05).” The redundant “HB” has been removed from the second comparison. The revised version has been updated in the manuscript. Page 8, 3.3. Serum Hormone Indicators and Relevant Gene Expression in the Ovary, and line 211.

Comments 22: Line 222: This is confusing “the highest MDA concentration in HB was 338.78 nmol/ml (P<0.05)”

Respond 22: Thank you very much for pointing this out. After careful rechecking, we found that there was a data entry error in the original table: the MDA value of 338.78 nmol/ml, which belongs to the HB, was mistakenly recorded under the HLA group. However, we would like to clarify that the description in the text-stating that the MDA value of HB was 338.78 nmol/ml (P < 0.05) is correct. We have now corrected the error in the table accordingly in the revised manuscript. We sincerely apologize for this oversight. Page 9, Table 7, and line 239.

Comments 23: Line 225: Review “and HLA and HB were significantly higher than that in HB 225 (P<0.05).”

Respond 23: Thank you for catching this mistake. We have corrected line 225 to: “GPx concentration in the CON was higher than in the HB and HL (P<0.05), the HB was lower than all other groups (P<0.05), and the HLA showed no significant difference compared with CON and HL (P>0.05).” The revised version has been updated in the manuscript. Page 9, 3.4. Serum Antioxidant Index and Relevant Gene Expression in the Liver, and line 233-236.

Comments 24: Line 248 – 249: Don't repeat values already presented in the tables.

Respond 24: Thank you for pointing this out. We have removed all redundant numerical values from the Results text. The revised version has been updated in the manuscript. Page 9-10, 3.5. Serum Immunoglobulin, Inflammatory Cytokines Indicators and Relevant Gene Expression in the Liver, and line 257-261.

Round 2

Reviewer 2 Report

Comments and Suggestions for Authors

The manuscript has been improved compared to the previous version and, in its current form, I consider it suitable for publication.